Elevational pattern of bird species richness and its causes along a central Himalaya gradient, China

Pan Xinyuan 1 2 3 4
Ding Zhifeng 3 4
Hu Yiming 2 3 4 5
Liang Jianchao 3 4
Wu Yongjie 6
Si Xingfeng 7
Guo Mingfang 3 4
Hu Huijian 13922339577@139.com 3 4
Jin Kun jk2002@caf.ac.cn 8
1 South China Botanical Garden, Chinese Academy of Sciences , Guangzhou , Guangdong , China
2 University of Chinese Academy of Sciences , Beijing , China
3 Guangdong Institute of Applied Biological Resources , Guangzhou , Guangdong , China
4 Guangdong Public Laboratory of Wild Animal Conservation and Utilization , Guangzhou , Guangdong , China
5 Key Laboratory of Animal Ecology and Conservation Biology, Institute of Zoology, Chinese Academy of Sciences , Beijing , China
6 Key Laboratory of Bio-resources and Eco-environment (Ministry of Education), College of Life Sciences, Sichuan University , Chengdu , Sichuan , China
7 College of Life Sciences, Zhejiang University , Hangzhou , Zhejiang , China
8 Research Institute of Forest Ecology Environment and Protection, Chinese Academy of Forestry , Beijing , China
Roberts David
Electronic publication date: 2016 Nov 2
Publication date: 2016
Volume: 4
Electronic Location ID: e2636
Received 2016 Jun 14; Accepted 2016 Sep 30
Copyright: ©2016 Pan et al.
Copyright year: 2016
Copyright holder: Pan et al.
License: This is an open access article distributed under the terms of the Creative Commons Attribution License, which permits unrestricted use, distribution, reproduction and adaptation in any medium and for any purpose provided that it is properly attributed. For attribution, the original author(s), title, publication source (PeerJ) and either DOI or URL of the article must be cited.
License URL: https://creativecommons.org/licenses/by/4.0/

Keywords: Bird species richness, Elevational pattern, Environmental factors, Spatial factors

Funding: National Natural Science Foundation of China 31400361 State forestry administration of the People’s Republic of China This research was supported by grants from the National Natural Science Foundation of China (No. 31400361), the state forestry administration of the People’s Republic of China “The special investigation of Himalayan Tahr population and habitat,” and the state forestry administration of the People’s Republic of China “The special investigation of Hanuman Langur population and habitat.” The funders had no role in study design, data collection and analysis, decision to publish, or preparation of the manuscript.

==============================
This study examines the relative importance of six variables: area, the mid-domain effect, temperature, precipitation, productivity, and habitat heterogeneity on elevational patterns of species richness for breeding birds along a central Himalaya gradient in the Gyirong Valley, the longest of five canyons in the Mount Qomolangma National Nature Reserve. We conducted field surveys in each of twelve elevational bands of 300 m between 1,800 and 5,400 m asl four times throughout the entire wet season. A total of 169 breeding bird species were recorded and most of the species (74%) were small-ranged. The species richness patterns of overall, large-ranged and small-ranged birds were all hump-shaped, but with peaks at different elevations. Large-ranged species and small-ranged species contributed equally to the overall richness pattern.

Based on the bivariate and multiple regression analyses, area and precipitation were not crucial factors in determining the species richness along this gradient. The mid-domain effect played an important role in shaping the richness pattern of large-ranged species. Temperature was negatively correlated with overall and large-ranged species but positively correlated with small-ranged species. Productivity was a strong explanatory factor among all the bird groups, and habitat heterogeneity played an important role in shaping the elevational richness patterns of overall and small-ranged species. Our results highlight the need to conserve primary forest and intact habitat in this area. Furthermore, we need to increase conservation efforts in this montane biodiversity hotspot in light of increasing anthropogenic activities and land use pressure.

Introduction

Geographical variation in biodiversity is one of the most conspicuous patterns in biology. Developing an understanding of this variation has been of interest to naturalists and biogeographers ever since Darwin and Wallace (Heaney, 2001) and has taken on new urgency since ongoing human activities may soon lead to the extinction of the majority of extant species (Dirzo & Raven, 2003; Wiens, 2011). Altitudinal gradients have received much attention in recent decades and have become increasingly popular for uncovering the underlying mechanisms of the spatial variation in diversity, owing to various advantages compared to latitudinal gradients (e.g., globally replicated gradients, smaller spatial scale and constancy in the ecological and historical context of the faunal assemblage) (McCain, 2009).

Traditionally, species richness has been considered to decrease monotonically with increasing elevation because of reduced temperatures and a consequent decrease in productivity (Rohde, 1992; Stevens, 1992). However, when standardized for area effects and sampling effort, different elevational patterns emerge, which can generally be grouped into four categories: a decrease in species richness with elevation, a low-elevation plateau followed by a decrease, a low-elevation plateau with a mid-elevation peak, and a unimodal mid-elevation peak (Rahbek, 1995; Rahbek, 2005). Numerous hypotheses have been proposed to explain both linear and humped relationships between species richness and altitude (Rahbek, 1997; Lomolino, 2001; McCain, 2007a; McCain, 2007b; Wu et al., 2013b; Rowe, Heaney & Rickart, 2015). Generally, spatial (e.g., area, mid-domain effect (MDE)) (Rahbek, 1997; Colwell, Rahbek & Gotelli, 2004) and environmental factors (e.g., climatic variables, productivity and habitat heterogeneity) (Nogués-Bravo et al., 2008; Sanders & Rahbek, 2012; Wu et al., 2013b) are the most frequently cited explanations for the variation in species richness.

Area has been found to explain a large proportion of the elevational species richness patterns (Rahbek, 1997), and subsequent studies have confirmed the role of area in shaping species richness patterns (Fu et al., 2004; Kattan & Franco, 2004; McCain, 2005; Williams et al., 2010). The MDE is defined as “the increasing overlap of species ranges towards the center of a shared, bounded domain owing to geometric boundary constraints” (Colwell & Hurtt, 1994; Colwell, Rahbek & Gotelli, 2004) and has been suggested to explain a substantial proportion of the elevational variation in richness (McCain, 2004; Cardelús, Colwell & Watkins, 2006; Kluge, Kessler & Dunn, 2006; Rowe, 2009; Wu et al., 2013a; Wu et al., 2013b).

The climate–richness relationship is widely supported worldwide (Currie et al., 2004). Climatic variables, often captured as the mean annual temperature (MAT, also referred to as ‘temperature’ below) and mean annual precipitation (MAP, also referred to as ‘precipitation’ below), are considered to be the most widely supported environmental predictors of biodiversity patterns (Hawkins et al., 2007; McCain, 2007b; Wu et al., 2013a; Wu et al., 2013b). Temperature and precipitation can influence biodiversity both directly through physiological tolerances and indirectly by affecting food resource availability (Currie et al., 2004; McCain, 2009; Rowe, Heaney & Rickart, 2015). The niche conservatism hypothesis proposes that most modern species originated when the majority of the Earth was experiencing tropical-like conditions and that these species tend to retain their ancestral ecological characteristics. Thus, there will be high species richness in regions characterized by warm temperatures and abundant rainfall (Wiens & Donoghue, 2004).

The energy-diversity hypothesis proposes that increased energy availability often results in more species rather than larger populations of existing species (Hutchinson, 1959; MacArthur, 1972; Wright, 1983). As a proxy for energy availability, primary productivity has often been estimated using the normalized difference vegetation index (NDVI). Previous studies on birds have reported a positive relationship between the NDVI and species richness (Hurlbert & Haskell, 2003; Lee et al., 2004; Hawkins, Diniz-Filho & Soeller, 2005; Koh, Lee & Lin, 2006; Wu et al., 2013b). The habitat heterogeneity (HH) hypothesis states that more habitat types or greater structural complexity in vegetation can provide more resources and may therefore support a larger number of species (MacArthur & MacArthur, 1961). HH appears to have been less frequently tested than climatic factors in previous studies, possibly because of difficulties associated with measuring this variable; for example, the relevant type of heterogeneity will depend on the taxa studied and on the scale of the study (Heaney, 2001; Rowe, 2009). However, when taken into account, a positive role of HH in shaping species richness patterns is often significant (Sánchez-Cordero, 2001; Hurlbert & Haskell, 2003; Hurlbert, 2004; Koh, Lee & Lin, 2006; Rowe, 2009).

Most studies of the determinants of geographic patterns in species richness have traditionally focused only on overall species richness patterns. However, geographic patterns in overall species richness are usually dominated by wide-ranging species because their larger number of distribution records has a disproportionate contribution to the species richness counts than do narrow-ranging species. The geographic range size could influence our understanding of what determines species richness (Jetz & Rahbek, 2002). Some studies have demonstrated that different patterns and processes determine the elevational diversity of large-ranged and small-ranged species (Cardelús, Colwell & Watkins, 2006; Kluge, Kessler & Dunn, 2006; Wu et al., 2013b).

Despite two centuries of investigation, the mechanisms shaping species richness patterns along elevational gradients remain controversial (Rahbek, 2005; Rowe, 2009). Comparing independent transects and searching for similarities and differences in patterns among transects and taxa in different climates and biogeographic regions can certainly improve our understanding of the mechanisms underlying biodiversity patterns (Grytnes & McCain, 2007). The Himalayas contain the highest mountains in the world, with a diverse range of eco-climate zones (Dobremez, 1976), and are one of the world’s biodiversity hotspots (Myers et al., 2000). Mountains with a broad range of elevation such as this have been considered ideal systems for elevational diversity studies (Acharya et al., 2011).

Previous studies in this region have focused on plants (Grytnes & Vetaas, 2002; Bhattarai & Vetaas, 2003; Oommen & Shanker, 2005; Bhattarai & Vetaas, 2006; Acharya, Vetaas & Birks, 2011) and reptiles (Chettri, Bhupathy & Acharya, 2010), revealing a predominant unimodal pattern. For the elevational pattern of birds, one study from the Nepal Himalayas found that species richness decreased with increasing elevation (Hunter & Yonzon, 1993). Two descriptive studies from the central Himalayas, China, showed hump-shaped richness patterns of overall bird species (Li et al., 2013; Hu, Jin & Tian, 2016). Acharya et al. (2011) found that bird species richness along an eastern Himalaya gradient peaked at mid-elevations and was significantly correlated with primary productivity and habitat variables. A study of songbirds in the eastern Himalayas demonstrated a hump-shaped elevational richness pattern and found that elevational distributions were well-explained by resource availability (Price et al., 2014). One study of birds in the western Himalayas showed a hump-shaped elevational richness pattern and a significant correlation between species richness and vegetation structure (Joshi & Bhatt, 2015). Considering that the mechanisms underlying elevational patterns vary even among gradients with similar biogeographic histories and fauna (Rowe, 2009), new, optimally designed elevational studies and integrative analyses of biodiversity along central Himalaya gradients in China are important for understanding these complex patterns and their underlying mechanisms (Grytnes & McCain, 2007; Wu et al., 2013a) and for the management and conservation of biodiversity.

In this study, we document the elevational species richness patterns of birds (using data obtained from a field survey in the Gyirong Valley, which is located on the southern slope of the central Himalayas, China) and assessed the ability of two spatial factors (area, MDE) and four environmental factors (MAT, MAP, NDVI and HH) to explain elevational patterns of bird species richness.

Materials and Methods

Study area

The Gyirong Valley (28°15′–29°0′N, 85°6′–85°41′E, Fig. 1) is the longest of the five canyons in the Mount Qomolangma National Nature Reserve and is characterized by an ecotone between the Oriental and Palearctic regions. This valley ranges from 1,680 to 5,770 m asl, with a complicated geological structure, varied geomorphologic types and rich biodiversity. The total area of this valley is 2,612 km2. Located in the subtropical monsoon climate zone, the valley is influenced by the warm, moist flow from the Indian Ocean and has distinct wet and dry seasons: the rainy season occurs from May to October, with the majority of rainfall occurring in July to September, and the dry season lasts from November to April.

Figure 1 Location of the study area.

The study area encompasses 12 sampling elevational bands. The numbers from 1 to 12 are the midpoints of transect lines distributed in the 12 elevational bands from the lowest elevation to the highest elevation (e.g., “1” was the midpoint of the transect lines distributed in the lowest elevational band).

There are five vegetation zones along the elevational gradient (Feng, Cai & Zheng, 1986; The Comprehensive Scientific Expedition to Qinghai-Xizang Plateau, Chinese Academy of Sciences, 1988): evergreen broadleaf forest (1,700–2,500 m asl), coniferous and broadleaf mixed forest (2,500–3,300 m asl), dark coniferous forest (3,300–3,900 m asl), shrub and grass (3,900–4,700 m asl) and alpine tundra with sparse grass (4,700–5,500 m asl).

Bird surveys

Prior to conducting field surveys, we obtained permits for the research from the Mount Qomolangma National Nature Reserve, and no bird was captured during the entire survey period. Field surveys of birds were conducted at 1,800–5,400 m asl and could not be performed at lower or higher elevations because of geographic restrictions. We divided the study area into 12 elevational bands of 300 m. Within each band, three transect lines that varied in length from 2,000 to 3,000 m were distributed to cover all the habitat types (Fig. 1). Because biased samples can affect the observed species richness pattern (Rahbek, 1995; Rahbek, 2005), the total length of all transect lines in each band was restricted to 7.5 km to ensure that the sampling effort was equally distributed across the gradient.

We recorded the presence and abundance of bird species together with information regarding their position with a handheld GPS (Magellan eXplorist 310) using standard line transect methods (Bibby et al., 2000). To increase the probability of detecting elusive or rare species, for all transect lines, bird surveys were carried out four times throughout the entire wet season (from May to June in 2012, August in 2012, from September to October in 2012, and from July to August in 2013). The surveys were conducted between 30 min after dawn and 11 AM (local time) and between 3 PM and 30 min before sunset; the surveys were not conducted at mid-day or during inclement weather owing to low bird activity. The taxonomic system used in this study followed Zheng (2011).

Species ranges

We used breeding birds (breeding birds were defined as those birds that breed in the study area, i.e., resident birds and summer visitors) for subsequent analyses, owing to a potential bias in the elevational range size associated with seasonal, long-distance migrants (McCain, 2009; Wu et al., 2013b). Species were assumed to occur within a band if they were observed within higher and lower elevational bands (Colwell & Hurtt, 1994; Colwell, Rahbek & Gotelli, 2004). This interpolation method is commonly used and has been widely regarded as valid in previous studies (Rahbek, 1997; Brehm, Colwell & Kluge, 2007; Wu et al., 2013b). In addition, it also avoids the underestimation of bird diversity owing to insufficient surveying of birds at both the temporal and spatial scale. The range size of each species was then transformed by “n × 300” m (“n” means the interpolated range of this species distributed over “n” elevational bands) to conduct the analyses described below.

Spatial factors

Area

We used GDEM 30-m digital elevation data from the International Scientific & Technical Data Mirror Site, Computer Network Information Center, Chinese Academy of Sciences (abbreviated as CNIC, CAS below; http://www.gscloud.cn/) to calculate the amount of three-dimensional surface area for each 300-m elevational band in ArcGIS 10.2 (ESRI, Redlands, CA, USA).

The mid-domain effect

We used RangeModel 5 (Colwell, 2008; http://purl.oclc.org/rangemodel) to randomize (without replacement) the empirical species ranges within the bounded domain to generate a predicted species richness pattern under geometric constraints (in the complete absence of any supposition of environmental gradients within the domain, see Colwell & Lees, 2000 for details). Predicted values and their 95% confidence intervals were computed for each 300-m band based on the mean of 5,000 simulations of the geometrically constrained null model.

Environmental factors

Mean annual temperature and mean annual precipitation

Fine-scale climatic datasets covering the entire planet from the WorldClim database (http://www.worldclim.org) are based on information from many meteorological stations, augmented by statistical extrapolations to regions without meteorological stations by the use of digital elevation models (Hijmans et al., 2005). Thus, corresponding digital maps with a horizontal grid spacing of 30 arc-seconds and including information on elevation, mean annual temperature, and mean annual precipitation were extracted from the WorldClim database (1950–2000). We obtained the values of temperature and precipitation in each 300-m band by averaging all grid cells within the band based on the elevational value of each grid cell in ArcGIS 10.2 (ESRI, Redlands, CA, USA).

Productivity

For the above-ground net primary productivity, we averaged the NDVI data for the Gyirong Valley from the Ministry of Environment Protection of the People’s Republic of China (http://www.zhb.gov.cn) for each elevational band over four years (2011–2014) using ERDAS IMAGINE 9.2 (ERDAS, Norcross, GA, USA).

Habitat heterogeneity

HH was summarized using the Shannon diversity index (the abundance of one identified habitat type = the area of that particular habitat type, richness = the number of different habitat types), which is commonly applied at the landscape scale (Turner & Gardner, 2015). We combined the GlobCover land cover data from CNIC, CAS (http://www.gscloud.cn/) and a 30-m digital elevation model (DEM) of the Gyirong Valley to calculate the area for each land-cover type in each 300-m elevational band using ArcGIS 10.2 (ESRI, Redlands, CA, USA). Twenty-two land-cover types are defined and primarily reflect the anthropogenic land use and the different types of forest, woodland, shrubland, and herbaceous communities.

Data analyses

To assess the effect of range size on the determinants of elevational patterns in species richness, we divided the overall species into two categories, the “large-ranged” category of species with ranges equal to or above the median size (elevational range size 1,800 m) and the “small-ranged” category of species with ranges below the median size (Wu et al., 2013a).

It is unlikely to detect all species in natural communities over limited time and space (Colwell & Coddington, 1994; Chao et al., 2005; Walther & Moore, 2005); thus, we used non-parametric estimators (Chao2 and Jackknife2) to compute the estimated species richness (Colwell & Coddington, 1994) using the statistical software program EstimateS 9.0 (Colwell, 2013; http://purl.oclc.org/estimates). Regression of the observed species richness against the estimated species richness was then performed to assess whether species diversity was sampled adequately for the elevational gradient.

Polynomial regressions were performed to clarify the elevational distribution pattern of interpolated species richness as a function of elevation along the gradient. We used the corrected Akaike information criterion (AICc) to compare the fits of first-order, second-order and third-order polynomial regressions, and smaller AICc values indicated a better fit. Before relating species richness to candidate explanatory variables (Area, MDE, MAT, MAP, NDVI, and HH), we used Spearman’s rank correlation to examine the relationships among the independent variables.

We performed simple ordinary least squares (OLS) regressions of the interpolated species richness for each species group (overall, large-ranged and small-ranged species) against each of the 6 candidate factors (Area, MDE, MAT, MAP, NDVI, and HH) to explore the role of individual factors in shaping elevational species richness patterns. To correct for spatial autocorrelation in the regression residuals, we calculated the effective number of degrees of freedom for each regression and reported adjusted P-values based on the effective degrees of freedom (Dutilleul, 1993).

We performed multiple regressions to explore the multivariate explanations for elevational patterns of bird species richness. For each species group (overall, large-ranged and small-ranged species), the best model was selected from the 63 models representing all possible combinations of 6 candidate explanatory variables (Area, MDE, MAT, MAP, NDVI, and HH), guided by the lowest AICc (Anderson, Burnham & White, 1998). However, there was sometimes nearly equivalent support for multiple models (i.e., nearly equal AICc or ΔAICcvalues, i.e., ΔAICc < 2, see Table S5); thus, we used a model-averaging approach to compare the selected best models and assess the relative importance of different drivers by standardized beta coefficients (Anderson & Burnham, 2002; Johnson & Omland, 2004).

The spatial autocorrelation in regression residuals and multicollinearity among explanatory variables could affect the credibility of the results and need to be taken into account (Diniz-Filho, Bini & Hawkins, 2003; Graham, 2003). However, in the case of a limited sample size, it is not feasible to apply spatial autoregressive analyses with six explanatory variables. Thus, no P-values were reported for the multiple regressions (Brehm, Colwell & Kluge, 2007). To reduce the multicollinearity in the model, we conducted multiple OLS models without Area, AET, and AEP because they are highly correlated with NDVI (Table 1). Only MDE, NDVI and HH were tested in the multiple OLS regressions for all species groups (overall, large-ranged and small-ranged species). Because the collinearity among explanatory variables cannot be confidently resolved with such a small sample size (Graham, 2003), we furthermore performed partial regression for the different species groups (overall, large-ranged and small-ranged species) with three variables (MDE, NDVI, HH) partitioned into a spatial variable (MDE) and environmental variables (NDVI, HH) to give a representative picture of the unique and shared contributions of the spatial variable and the environmental variables to the richness patterns.

Table 1 Spearman correlation coefficients for the six selected factors.

	Area	MAT	MAP	NDVI	HH	
Area						
MAT	−0.993*					
MAP	−0.993*	0.986*				
NDVI	−0.993*	0.986*	1*			
HH	0.014	−0.021	0	0		
MDE	0.126	−0.14	−0.112	−0.112	0.93*	
Notes.

* P < 0.01.

MAT mean annual temperature

MAP mean annual precipitation

NDVI normalized difference vegetation index

HH habitat heterogeneity

MDE the mid-domain effect

Polynomial regressions were performed in PAST 2.17 (Hammer, Harper & Ryan, 2001; http://folk.uio.no/ohammer/past/). Spearman correlation analyses and bivariate and multiple regression analyses were performed in SAM 4.0 (Rangel, Diniz-Filho & Bini, 2010; http://www.ecoevol.ufg.br/sam).

Results

Elevational diversity patterns

Area increases monotonically with elevation, and MAT, MAP, and NDVI all decrease with elevation, whereas HH shows a hump-shaped pattern along the elevation gradient (Fig. 2).

Figure 2 Elevational patterns of (A) area, (B) temperature, (C) precipitation, (D) NDVI (normalized difference vegetation index) and (E) habitat heterogeneity.

A total of 169 breeding bird species were recorded in the Gyirong Valley, belonging to 11 orders, 41 families and 100 genera (Table S1). The regression of the observed species richness against the estimated species richness (Table S2) (Chao2, r2 = 0.914, P < 0.01; Jackknife2, r2 = 0.977, P < 0.01) indicated that the sampling was adequate to accurately characterize the species richness patterns along the elevational gradient.

The interpolated species richness showed hump-shaped patterns along the elevational gradient (Fig. 3 and Table S3). The species richness of overall birds peaked at 2,700–3,000 m asl. The richness of large-ranged species peaked at 3,300–3,600 m asl, whereas that of small-ranged species had two peaks, with the larger peak occurring at 2,700–3,000 m asl and the smaller peak occurring at 3,600–3,900 m asl (Fig. 3).

Figure 3 Elevational patterns of interpolated species richness.

(A): large-ranged species; (B): small-ranged species; (C): overall species. Predicted species richness (gray solid lines) under the assumption of random range placement (MDE, the mid-domain effect) and the upper and lower 95% confidence interval simulation limits (gray dotted lines) are shown in the figure.

Explanatory factors

Spatial factors

Area was significantly correlated with the species richness of each species group in the simple OLS regressions (when spatial autocorrelation was not taken into account, Table 2). The beta coefficient for best-fit models (with the lowest AICc values) showed that not all of the species groups were correlated with area (Table 3). According to the model-averaging analyses, area was negatively correlated with all bird groups (Table S4).

Table 2 Simple ordinary least squares (OLS) regression analyses of interpolated species richness against six factors for different species groups.

Species groups		Six factors	
		Area	MAT	MAP	NDVI	HH	MDE	
Overall species	r2	0.632(-)	0.386	0.169	0.654	0.684	0.446	
	P	0.002	0.031	0.185	0.001	0.011	0.014	
	Padj	0.207	0.237	0.447	0.261	0.111	0.177	
Large-ranged species	r2	0.741(-)	0.46	0.234	0.763	0.307	0.344	
	P	<0.001	0.012	0.097	<0.001	0.052	0.037	
	Padj	0.452	0.308	0.397	0.498	0.243	0.292	
Small-ranged species	r2	0.467(-)	0.28	0.098	0.486	0.611	0.534	
	P	0.011	0.065	0.301	0.009	0.002	0.005	
	Padj	0.156	0.282	0.561	0.15	0.035	0.069	
Notes.

MAT mean annual temperature

MAP mean annual precipitation

NDVI normalized difference vegetation index

HH habitat heterogeneity

MDE the mid-domain effect

Padj is the adjusted P-value for r2 based on the adjustment of the degrees of freedom to account for spatial autocorrelation using Dutilleul’s (1993) method. Bold numbers indicate significant r2 values (P < 0.05, Padj < 0.05). Negative relationships are indicated by (-).

Table 3 Parameter estimates for the best-fit multiple regression models.

Species groups	Standard coefficient of the best model	
	Area	MAT	MAP	NDVI	HH	MDE	radj2	
Overall birds				0.728	0.584		0.988	
Large-ranged species		−0.562		1.352		0.328	0.991	
Small-ranged species				0.601	0.699		0.962	
Notes.

MAT mean annual temperature

MAP mean annual precipitation

NDVI normalized difference vegetation index

HH habitat heterogeneity

MDE the mid-domain effect

radj2 is the adjusted r2 value for multiple regressions. For each species group (overall, large-ranged and small-ranged species), the best model was selected from the 63 models obtained by forming all possible combinations of six variables (Area, MAT, MAP, NDVI, HH, MDE), guided by the lowest corrected Akaike information criterion value (AICc). All 63 models with their Δ AICc and AICc weights for all species groups are reported in Table S5.

The MDE was significantly correlated with the species richness of all species groups in the simple OLS regressions (when spatial autocorrelation was not taken into account, Table 2). In the best-model-selection and the model-averaging analyses, MDE was only identified to be a weak predictor of the richness pattern of large-ranged species (Table 3 and Table S4). The MDE played an important role in shaping the richness patterns of large-ranged species in the multiple OLS regressions with the three selected factors (to minimize multicollinearity, Table 4). In the partial regression analyses, the unique contribution of MDE to the richness patterns was weak for all the species groups (Table 5).

Table 4 Multiple ordinary least squares (OLS) regression of interpolated species richness against three selected factors for different species groups.

Species groups	NDVI	HH	MDE	r2	P	
Overall birds	0.728	0.568	0.017	0.989	<0.001	
Large-ranged species	0.804	0.6	0.408	0.974	<0.001	
Small-ranged species	0.599	1.017	−0.328	0.972	<0.001	
Notes.

MAT mean annual temperature

MAP mean annual precipitation

NDVI normalized difference vegetation index

HH habitat heterogeneity

MDE the mid-domain effect

Bold numbers indicate the parameters for each multiple regression model that were significant at P < 0.05.

Table 5 Partial regression for species richness of all bird groups with three selected factors partitioned into spatial (MDE, mid-domain effect) and environmental factors (NDVI, normalized difference vegetation index; HH, habitat heterogeneity).

Species groups	a	b	c	a + b	b + c	d	
Overall birds	<0.001	0.446	0.543	0.446	0.989	0.011	
Large-ranged species	0.02	0.324	0.631	0.344	0.955	0.026	
Small-ranged species	0.007	0.528	0.438	0.534	0.966	0.028	
Notes.

‘a’ and ‘c’ represent the unique contributions of spatial and environmental factors; ‘b’ is the shared contribution; ‘d’ is the unexplained variation; ‘a + b’ is the total contribution of the spatial factor to bird species richness; ‘b + c’ is the total contribution of environmental factors to bird species richness.

Environmental factors

Most of the environmental factors, except for precipitation, were significantly correlated with the species richness in each species group in the simple OLS regressions (when spatial autocorrelation was not taken into account, Table 2).

The beta coefficient for best-fit models showed that temperature was only negatively correlated with large-ranged species (Table 3). In the model-averaging analyses, temperature was negatively correlated with overall and large-ranged species but positively correlated with small-ranged species (Table S4).

The beta coefficient for best-fit models showed that not all of the species groups were correlated with precipitation (Table 3). Precipitation was an important explanatory factor for small-ranged species in the model-averaging analyses (Table S4).

The results of the best-model-selection, model averaging and multiple OLS regressions with the three selected factors (to minimize multicollinearity) all showed that productivity was a strong explanatory factor among all the bird groups and that HH played an important role in shaping the elevational richness patterns of overall and small-ranged species. Productivity best explained the species richness patterns of overall and large-ranged birds, whereas HH best explained the richness pattern of small-ranged birds (Tables 3, 4 and Table S4).

In the partial regression analyses, the unique contributions of the environmental variables (NDVI and HH) to the richness patterns was stronger than that of MDE for all the species groups, whereas the shared contribution varied among the species groups (32.4%–52.8%, Table 5).

Discussion

Elevational diversity patterns

Our finding that the overall bird species richness peaked at intermediate elevations (2,700–3,000 m asl, Fig. 3) was consistent with most previous studies on mammals (Hu et al., 2014), birds (Acharya et al., 2011; Joshi & Bhatt, 2015), reptiles (Chettri, Bhupathy & Acharya, 2010), and plants (Grytnes & Vetaas, 2002; Bhattarai, Vetaas & Grytnes, 2004; Acharya, Vetaas & Birks, 2011). The peak in richness is followed by a plateau at elevations of between 3,000 m and 3,900 m asl (Fig. 3). One possible explanation for this pattern is that the area between 3,100 and 4,000 m asl represents a transition zone between the Oriental and Palearctic regions (Tibetan Scientific Expedition of Chinese Academy of Sciences, 1974; Li et al., 2013), and such a pattern might provide new evidence for the hypothesis that ecotones between different faunas harbor more species (Brown, 1995).

Fu et al. (2006) and Brehm, Colwell & Kluge (2007) found that large-ranged species contribute more to overall richness patterns than small-ranged species, but our results were not consistent with this observation because the correlation coefficient between large-ranged and overall species was equal to that between small-ranged and overall species (both r2 = 0.942, P < 0.01). The equal contributions of groups with different range sizes to the overall richness pattern was perhaps due to the larger number of small-ranged species relative to that of large-ranged species (125 versus 44).

Spatial factors

The most recent synthetic analysis at a global scale found no consistent support for the influence of area on bird species richness along elevational gradients (McCain, 2009). The variability in area effects was largely attributed to the shape of the richness-elevation relationship and the area-elevation relationship (McCain, 2007a). A strong richness-area relationship was expected when both variables show concordant patterns along the elevational gradient. In this study, area increased monotonically with elevation (which was different from the patterns commonly observed for mountains), making Gyirong Valley an exceptional test system for evaluating the importance of area on species richness patterns. A negative relationship between species richness and area was demonstrated for all the groups in both individual regression analyses (Table 2) and multi-model analyses (Table S4), and not all the species groups were correlated with area in the best models (Table 3), indicating that area was not a crucial factor in determining species richness in the present study. Furthermore, Rahbek (1997) argued that it is not ground area per se that determines species richness but rather the volume of available habitat. In the Gyirong Valley, although the highest elevational bands harbored a larger area, the climate was cold and arid, resulting in unsuitable habitat for most bird species.

The MDE was an important variable in explaining species richness patterns along the elevational gradient based on the individual regression analyses (Table 2), but when considered in combination with other candidate factors using multi-model inference, the MDE appeared to be a weak predictor for all the species groups except for large-ranged species (Tables 3, 4 and Table S4). The MDE should be more pronounced when larger ranges are considered (Colwell & Lees, 2000; Colwell, Rahbek & Gotelli, 2004). In our study, the explanatory power of the MDE was also found to be stronger for large-ranged species than for small-ranged species in the multiple regressions (Tables 3, 4 and Table S4). The weaker explanatory power of the MDE in the species richness of the other bird groups was in some ways due to collinearity among explanatory variables because the variation shared with the MDE was “captured” by environmental factors (Tables 4 and 5). Our results highlight that both individual and multiple regression are important to develop a deeper understanding of the mechanisms underlying diversity patterns. In the present case, the species richness of all the bird groups peaked at lower elevations than predicted by the MDE null model, suggesting that other factors had modified the influence of the MDE.

Environmental factors

Acharya et al. (2011) found a strong correlation between bird species and climate variables in the eastern Himalayas. In our study, the climate-richness relationship was supported, whereas the explanatory power of temperature and precipitation varied across the different species groups (Tables 2, 3 and Table S4). The prediction of niche conservatism proposed that the warm, wet climates would harbor the most species on mountains (Wiens & Donoghue, 2004; Wiens & Graham, 2005); our results did not coincide with this prediction because the species richness did not decrease with decreasing temperature and precipitation along the elevational gradient. However, the climate data were not generated locally in this study because there were no meteorological stations in the Gyirong Valley. Considering the complex topography and various microhabitats on mountains, more high-quality, small-scaled and long-term data on climate factors such as rainfall, humidity, and cloud cover need to be collected along elevational gradients to more accurately generate climatic models.

Productivity was strongly correlated with the species richness of all bird groups (Table 3 and Table S4). In the multiple OLS regression excluding Area, AET, and AEP to reduce multicollinearity, productivity played a statistically more explanatory role in shaping the richness patterns of all bird groups (Table 4). Our results added support for the utility of satellite-derived vegetation indices as proxies of productivity and revealed new evidence for the energy-diversity hypothesis.

HH was identified as an important predictor of the species-elevation relationship for all the bird groups except for large-ranged birds in the present study. The weak explanatory power of HH in explaining the richness patterns of large-ranged birds might be attributed to two issues: (1) the “large-ranged” birds were distributed more widely across the elevational gradient; thus, they were more adaptable and could adapt to more habitat types (White & Bennett, 2015) and tended to be habitat generalists in this area (relative to the “small-ranged” birds). The large number of habitat generalists might result in incongruence between habitat heterogeneity and species richness (Rowe, Heaney & Rickart, 2015); (2) the measures of habitat diversity used so far failed to capture critical microhabitat differences in composition or structural complexity that may influence the diversity of large-ranged birds. Our results call for caution when assessing the role of HH in shaping species richness patterns, especially for those groups of taxa that contain a large number of species, and the measure used should adequately reflect habitat use by the species groups studied.

Biodiversity conservation

The Gyirong Valley harbors 47% of the bird species recorded in the Mount Qomolangma National Nature Reserve, based on our field survey and the research of Li et al. (2013). Most of the breeding birds (74%) in the Gyirong Valley are small-ranged (i.e., elevational range size of below 1,800 m). Species with smaller elevational ranges are at a greater risk of extinction than species with larger elevational ranges (White & Bennett, 2015), and small-ranged species on mountains would be more threatened under global warming (Colwell et al., 2008), highlighting the need for increased conservation efforts in this area. The species richness of overall birds peaked at mid-elevations; however, the mid-elevational area is influenced by strong anthropogenic activities and land use pressures (e.g., grazing and habitat conversion, according to our observations during the field survey). Anthropogenic habitat alterations and shifts in land use patterns could exacerbate the challenges of global warming faced by montane birds (Colwell et al., 2008; McCain & Colwell, 2011). In our study, the species richness of overall birds was positively correlated with productivity and habitat heterogeneity, indicating that the existing primary forest in this valley is important for biodiversity conservation and that changes in land use should avoid reducing the availability and connectivity of suitable habitats along the gradient. In the present case, our knowledge of anthropogenic threats is still limited; thus, long-term monitoring and applied research are needed in this montane biodiversity hotspot to provide more valuable insights for biodiversity conservation.

Conclusions

The species richness of all the bird groups in the Gyirong Valley of the central Himalayas peaked at mid-elevations, and the different species groups showed different richness patterns along the elevational gradient. No single factor or suite of factors could explain the species richness patterns across all the bird groups. The important roles of productivity and HH in shaping the elevational richness patterns of most bird species groups highlights the need to conserve intact habitat in this montane biodiversity hotspot.

Supplemental Information

Table S1 Species checklists of all birds recorded over survey period in the Gyirong Valley

‘1, 2, 3, … 12’ were the twelve elevational bands distributed from the lowest elevation to the highest elevation along the gradient.

Click here for additional data file.

Table S2 Observed, estimated (Chao2, Jackknife2) and interpolated richness of birds in each elevational band in the Giyrong Valley

Click here for additional data file.

Table S3 Polynomial regressions of the interpolated species richness patterns along the elevational gradients for all the species groups

Click here for additional data file.

Table S4 Parameter estimates averaged across 63 ordinary least squares (OLS) models

Click here for additional data file.

Table S5 Supplemental table S5

Click here for additional data file.

We thank numerous graduates in our group for the bird surveys in the field. We also thank Dr. Zhixin Zhou and Dr. Daoying Lan for providing extensive comments on the draft manuscript.

Additional Information and Declarations

Competing Interests

Author Contributions

Field Study Permissions

Data Availability

The authors declare there are no competing interests.

Xinyuan Pan conceived and designed the experiments, performed the experiments, analyzed the data, wrote the paper, prepared figures and/or tables.

Zhifeng Ding conceived and designed the experiments, contributed reagents/materials/analysis tools, wrote the paper.

Yiming Hu and Jianchao Liang performed the experiments, analyzed the data, prepared figures and/or tables.

Yongjie Wu and Mingfang Guo wrote the paper, reviewed drafts of the paper.

Xingfeng Si analyzed the data, prepared figures and/or tables.

Huijian Hu conceived and designed the experiments, performed the experiments, contributed reagents/materials/analysis tools, reviewed drafts of the paper.

Kun Jin contributed reagents/materials/analysis tools, reviewed drafts of the paper.

The following information was supplied relating to field study approvals (i.e., approving body and any reference numbers):

Prior to field survey, we obtained the permits for the research from the Mount Qomolangma National Nature Reserve, and no bird was captured during the whole survey period.

The following information was supplied regarding data availability:

The raw data has been supplied as Supplementary Files.

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
