# Peer review of "Elevational pattern of bird species richness and its causes along a central Himalaya gradient, China"

_PeerJ, doi:10.7717/peerj.2636_

## Round 0.1 · original submission · Major Revisions

This is potentially an interesting paper but lacks clarity. For it to be accepted it requires some work in terms of making the hypotheses, structure of the analysis and discussion clearer.

Reviewer 1 ·

Basic reporting

In general, the manuscript fails to meet the standard of clarity required for publication. A number of specific comments are addressed below referring to areas of the text that were unclear and require improvement. However, this is not an exhaustive list, and there were a number of times that sentences were not written using clear unambiguous language.

Edits to the text:
Commas are misused throughout the text. A number of dependent clauses are incorrectly separated by commas. See line 50: “decades, and have”; this comma is unnecessary. I would also suggest sentences be simplified for clarity. The first paragraph (lines 45-54) contains only two sentences and could be divided into a number of simpler sentences. Especially in the Abstract and Introduction, the writing should be simple enough to effectively outline the argument of the paper. Confusion in the first paragraphs has a cascading negative impact on the clarity of the entire paper. I urge the authors to consider shortening and dividing long sentences for clarity.

Line 56: A statement regarding the cause of monotonic species decline would be helpful.

Line 61: Paragraph break is not needed.

Line 63: “spatial” incorrectly capitalized. Also, MDE has only been used in the abstract. I suggest “mid-domain effect (MDE)” here and MDE thereafter.

Line 66: I suggest dividing these citations between spatial and environmental (e.g., “spatial (citations) and environmental (citations) factors are the most…”)

Line 71: There is an open quotation here with no close quotation.

Line 85: “Habitat heterogeneity” should be used here as it is the first mention in the main text.

Line 92: I don’t agree that these mechanisms are “poorly known”. Perhaps this is poor word choice. The exact causes remain unresolved, but as the authors document, there are a number of studies indicating support for a number of hypotheses which are well studied.

Line 94: The comma used here should be replaced with “and”.

Line 98: “Mountain” should be plural.

Line 103: Considering the authors cite 5 papers referring to studies of bird species richness patterns in the Himalayas, I would argue that this conflicts with the suggestion that these patterns are “rarely reported”. Himalayan biodiversity patterns are arguably poorly known for most taxa – the rarity of these studies is not a feature exclusive to the study of avian diversity.

Lines 107-115: This is the primary opportunity the authors have to explicitly detail the findings of those past studies of Himalayan bird diversity. None of those findings are mentioned in detail. Perhaps too much of this introduction is spent on explicating the generalities of elevational richness patterns and their study, but very little of the introduction refers specifically to what is already known about Himalayan bird diversity along elevational gradients. I suggest greater balance in the introduction between generality and specific results. Specifically, in this section, some rewriting is required as the grammar and sentence structure is unclear.

Line 118: What does “grandest” refer to here?

Line 120: Sentence is in past tense and the next is in present. Present tense seems most appropriate.

Line 127: I’m not sure this is the proper way to cite this report.

Figure 1: The smaller map of China is not helpful. Perhaps showing more detail of the regional setting of the study site would be better. For instance, it is not clear from these maps how the study site is set within the Himalayan range. It is important for this regional context to be clear. It is also not clear what the red boundary delineates in the detailed map.

Line 446: Gaston misspelled

Experimental design

While the authors address a number of hypotheses regarding species richness along elevational gradients, the overall clarity of the argument for doing so is muddled. I urge the authors to consider a more structured framework for presenting these hypotheses and apply that framework to the presentation of the results. Specific comments regarding the reporting of the methods sections are included below.

Line 134 – 150: It is not exactly clear how many times each transect was surveyed. Is it that each transect was survey 4 times? Line 145 refers to “bird samplings” but it is not clear what unit of study this refers to. Also, how was species presence identified? By site or sound or both? How was abundance determined if so? These details should be included. Also why not include the location of the transects in Figure 1?

Line 158: The empirical species ranges have not been discussed until now. In the previous section about bird surveys, a statement should be included about how the presences from the surveys were converted into species ranges. The first paragraph under “Data analyses” should be moved to the first section of the methods.

Line 173: Wiens and Graham 2005 is not the appropriate citation here. The correct citation is: Hijmans, R.J., S.E. Cameron, J.L. Parra, P.G. Jones and A. Jarvis, 2005. Very high resolution interpolated climate surfaces for global land areas. International Journal of Climatology 25: 1965-1978.

Line 182: This ftp address is not an appropriate citation.

Line 193: It is not clear how the authors distinguished between “breeding birds” and other classifications in their data.

Line 200: Are these not “elevational” patterns?

Line 205: Chao 2 and Jackknife 2 references should be added.

Line 209: The authors should include an explicit statement as to what is being modeled here. I assume that when the authors say “Polynomial regressions were performed to clarify the elevational distribution pattern of interpolated species richness,” they mean “species richness in each 300m band was regressed against spatial and environmental factors in order to examine how these factors shape species richness across the elevational gradient”. While this is my interpretation it is worth including an explicit statement to this effect.

Validity of the findings

In general, the results are not clearly reported. The main problem is that there are many different analyses, and they are not presented within an organized framework. For instance, it would help if the results were organized according to the hypotheses they are meant to address. Instead, the results are organized according to the tests that were performed, with a number of analyses addressing a number of hypotheses simultaneously. The subsection “Explanatory factors” is too general to help organize these various results for the reader. More specific subheadings would help.

Line 254: Is it correct to say that there was “no significant difference” between the values? The regression of one value against another merely addresses the relationship (causal or correlative) between the values. Perhaps a small difference in the language used, but looking at the supplementary table it is clear that the estimators predict more species than are observed. It may be more clear to say that the tests indicate that the sampling is adequate to accurately characterize the species richness patterns along the elevations gradient.

Line 268: It is not clear that Table 2 addresses each predictor in a separate model. Also perhaps Table 2 would be more clear if the explanatory variables were in the columns and the rows were divided into the sections by groups.

Line 271: Is it correct that Table 2 reports the R2 values? These appear to be the beta values, otherwise negative R2 values should just be reported as 0.

Line 277: Because there are a number of models being addressed, it would be helpful to say what “best fit models” are being discussed here. I assume these are multiple regressions taking all of the 6 factors into account.

Line 282: A model selection supplement is essential to interpret the AICc values. As reported, the AICc values are not helpful.

Line 287: It is not clear why the authors include a model averaging approach. Why model average across all 63 different models? Model averaging would be helpful if the AICc values indicated a number of “best fit” models within a given threshold (a common approach being those models with a difference of an AICc value <2), but without the authors reporting these values then the model averaging approach is not clearly motivated.

Line 357: Are the authors sure that the WorldClim data are not entirely interpolated across the study area? I am not aware of many climate stations in the region, and, if they are absent, here the authors may just mention that these climate data were not generated locally.

Additional comments

The authors provide a detailed account of the species richness patterns along elevations in an understudied but highly diverse region of the world. The data and the deliberate and organized collection of those data are an important and commendable contribution.

Overall, the submission suffers most from a lack of clarity. Secondary to that, the results need to be checked for their accuracy. Ultimately, after corrections have been made to the body of the text, I suggest the authors perhaps re-write the abstract.

Reviewer 2 ·

Basic reporting

Although the writing style is generally coherent and, overall, the structure flows well, there are a number of spelling mistakes/missing/additional words (I’ve highlighted some of the key issues). Some of the sentences are awkwardly worded. I therefore strongly recommend a thorough proof-reading of the article by a native English speaker.

Quite a number of abbreviations are used throughout this paper. Generally you define them on first mention; however, HH & MDE are only defined in the abstract (they need to also be defined on first mention in the main text). I also recommend not using abbreviations in sub-section titles.

Lines 79-81: bit simplistic, but you should cite a primary source(s) that has shown this.

Lines 81-90: explanation missing for how NDVI/HH affects bird diversity patterns. Also no mention of relationship direction.

Lines 91:92: why do you think this is?

Lines 103-106: Were no underlying processes investigated/suggested in these two studies?

Lines 98-99 & 107-111 & 113 & 198-199 & 212-213 & 231-232 & 342-344 & 361-363: Awkward sentence structure – please re-write.

There appears to be no reference to endemism in your study, which I think is an omission. Are any of the bird species endemic to your study area? A number of studies suggest that altitudinal patterns of endemic species diversity differ considerably to that of total species diversity patterns. For example, some studies have found endemic diversity to increase with altitude.

Line 118: colloquial language used.

Line 120: Past tense used for some reason – please correct.

More clarity needed in methods section regarding exactly what bivariate and multivariate analyses you did and for which variables.

For relevant tables, delete “Negative relationships are indicated by ‘-‘” – not necessary.

Line 242: replace “simple” with “bivariate”.

Appendix 1: what does the “endemic” column refer to? Endemic to what?

Experimental design

How large an area does your study site cover – I could not find this information in your study area section.

Line 137: what exactly is meant here by “geographic restriction”?

Line 138: how far apart were each of the three transect lines within a given elevational band? In other, words were they far enough apart so as to be considered independent?

Lines 144-147: to clarify, were each of the three transect routes per elevational band walked four times?

Line 185: A sentence needed on exactly what this involved; i.e. determining relative area of each identified habitat type?

Lines 193-194: How did your study deal with altitudinal migrants?

Lines 199-202: So what was your actual cut-off range size for separating “large-ranged” and “small-ranged” species? Where did you obtain geographical range size data from? In addition, no discussion is provided in your introduction regarding why you would be interested in looking at these two groups with respect to your study – needs to be explicitly mentioned; i.e. because large-ranged species can dominate diversity patterns.

Line 213: What was your reasoning for using Spearman’s rank correlation here? I assume the small sample size, and the data not being parametric?

Line 215: I do question the validity of performing regression-based analyses when you only have 12 data points (i.e. elevational bands) for your independent variable.

Line 220: what is meant here by “simple variables”?

Validity of the findings

Line 269: what is meant here by the term “different groups” or “species groups” (Line 277)? Bit misleading if you’re referring to large- and small-ranged birds, as the terminology used seems to instead infer either taxonomical or functional groups.

Table 2: it is interesting to see that there’s a negative relationship between species richness and area – this is the converse of what is “normally” found – no discussion in your discussion section regarding this. Also, in Table 3 – you find no area relationship.

Table 3: to confirm, all six predictors were not added into the same model here? Otherwise the model would be confounded due to multicollinearity.

Lines 288-289: not sure I understand your reasoning behind omitting area as an explanatory factor.

Line 302: what do these numbers refer to? Units needed.

Line 328: change to “species richness” or “species diversity”.

Line 328-330: not sure I follow here, just because a negative relationship was found between species richness and area doesn’t mean it’s not valid!

Lines 351-353: here you mention relationships and predictions which were not set-up in your introduction – this is needed.

Lines 368-370: not clear what is meant here – more and clearer detail needed.

Some of the statements made within both the introduction and discussion seem a bit “empty”. For example, [lines 45-46] why is it critical for biodiversity conservation to understand biogeographic patterns of species richness? You also mention that your study provides “valuable insights for conservation biologists” [lines 27-28] and how your “results highlight the conservation necessity of primary forest/intact habitat in this montane biodiversity hotspot” [lines 41-43], yet you do not discuss how [see also line 111 & 380-382]. This needs to be explicitly discussed in your discussion section. Related to this, I am surprised to see no reference to how anthropogenic threat-types and extinction risk varies with altitude; e.g. see White, R.L. & Bennett, P.M. (2015) Global elevational distribution and extinction risk in birds. PLOS ONE, 10, e0121849. What are the implications of your study, and how could it be expanded upon in the future?

Overall, I think the discussion section requires further integration of ecology (both ecological knowledge of the area and species, and also existing theory).

Additional comments

Overall, this is an interesting paper, but I think it is underdeveloped and too superficial in parts. Although there are now a number of published studies regarding (avian) altitudinal gradients of diversity, I appreciate the fact that this paper actually explores a number of potential underlying processes to the patterns found. Data analyses applied seems comprehensive yet I do question how robust it is. I have concerns about the small sample size (i.e. 12 data points for the independent variable of altitude). Both the introduction and discussion require considerable expanding upon. For example, the introduction lacks a priori hypotheses and does not contain all of the relevant existing theories surrounding this subject area. The discussion is currently too brief and descriptive – lacking a critical element. In addition, although the conservation value of this study is hinted at, no explicit discussion is provided here, which is a shame.

Annotated reviews are not available for download in order to protect the identity of reviewers who chose to remain anonymous.

---

## Round 0.2 · accepted · Accept

I had hoped to get an additional round of reviewers comments but unfortunately they were not available and I do not like send it out to reviewers who have not seen the first version of the manuscript. Having read the manuscript it has been improved significantly in terms of it clarity. I am therefore happy to accept it for publication. However I would suggest that during the proof stage to take another look at the abstract as it is a little long and it lack an opening couple of sentences introducing the paper. Remember this is what people read before deciding to look at the rest of the paper.